# Cellular Effects of Silver Nanoparticle Suspensions on Lung Epithelial Cells and Macrophages

**Kaori Shimizu [1,2], Shosaku Kashiwada [2] and Masanori Horie [1,\***

1 Health and Medical Research Institute, National Institute of Advanced Industrial Science and Technology (AIST), 2217-14 Hayashi-cho, Takamatsu 761-0301, Japan; kaori.shimizu9110@gmail.com

2 Graduate School of Life Sciences, Toyo University, 1-1-1 Izumino, Itakura 374-0193, Japan; kashiwada@toyo.jp

\* Correspondence: masa-horie@aist.go.jp; Tel.: +81-87-869-4197

**Abstract:** Background: Silver nanoparticles (AgNPs) are used in industrial applications as catalysts, sanitary materials, and health supplements. Generally, AgNPs have shown cytotoxicity such as cell membrane damage. However, the mechanisms of their toxicity have not been completely elucidated. Methods: The cellular effects (cell viability, induction of chemokine and cellular oxidative stress) of two AgNP water suspensions (AgNP-A for cosmetic application and AgNP-B for industrial application) on epithelial-like A549 cells and macrophage-like differentiated THP-1 (dTHP-1) cells were examined. Results: AgNPs caused enhancement of IL-8 expression and oxidative stress. The cellular uptake of AgNP-A cells was observed. However, the cellular uptake of AgNP-B into A549 cells was hardly observed. Moreover, the intracellular Ag level was increased by AgNP suspensions exposure. Cell viability was not affected by AgNP suspensions exposure. Conclusions: AgNPs induce chemokine expression and cellular oxidative stress on culture cells. The intracellular Ag level may be important for these cellular effects.

**Keywords:** silver; nanoparticle; cytotoxicity; interleukin-8; silver nitrate

## 1. Introduction

Nanoparticles are characterized by a diameter of 1–100 nm [1]. Many types of metal and metal oxide nanoparticles are currently being produced. Among these, silver nanoparticles (AgNPs) are particularly important because they are utilized not only for industrial applications such as catalysis, but also for familiar materials used in everyday life. AgNPs exhibit antimicrobial activity [2], and therefore they are widely used in various sanitary materials, such as antimicrobial textiles and deodorants [3–5]. In addition, AgNPs are utilized for wound-healing purposes as they inhibit microbial growth [6].

Compared with other metal or metal oxide nanoparticles, AgNPs are characterized by their antimicrobial activity. The mechanisms underlying the antimicrobial activity of AgNPs have been suggested: (1) the nanoparticles adhere to and accumulate on the cell wall, changing the cell structure or causing the formation of pits, thereby leading to cell death; (2) the nanoparticles generate reactive oxygen species (ROS); (3) the nanoparticles release silver ions, which exert bactericidal activity [2,7–9]. The released silver ions bind to thiol protein groups and disrupt bacterial metabolism by inhibiting enzyme activity [10]. Furthermore, AgNPs damage nucleic acids directly or through ROS generation [9]. These antimicrobial properties of AgNPs also indicate their potential cytotoxic activity against mammalian cells.

The cytotoxicity of AgNPs to mammalian cells has been reported [11]. For example, exposure to AgNPs leads to cell membrane damage and apoptosis in A549 human lung carcinoma-derived cells [12]. Moreover, AgNPs induce oxidative stress and DNA injury in normal human bronchial epithelial (BEAS-2B) cells [13]. Bioengineered AgNPs showed not only antimicrobial activity, but also cellular effects, such as a decrease of cell viability

and cellular oxidative stress induction in A549 cells [14]. Another biologically synthesized AgNP caused a decrease of cell viability [15]. In many cases, these cytotoxic activities against mammalian cells share similar mechanisms and common molecular events with antimicrobial activities. In addition to these antimicrobial mechanisms, cellular particle uptake is an important factor in the cytotoxicity of AgNPs to mammalian cells. Several metal oxide nanoparticles exhibit stronger cytotoxicity than their water-soluble compounds owing to the cellular uptake and subsequent intracellular release of metal ions [16–18]. For example, the cytotoxicity of nickel oxide nanoparticles is more potent than that of nickel chloride [16]. Moreover, ZnO and CuO nanoparticles release $Zn^{2+}$ and $Cu^{2+}$ extracellularly and intracellularly, and these metal ions subsequently cause cytotoxicity [19,20].

Phagocytes, such as macrophages and neutrophils, have a higher phagocytic capacity than epithelial cells. Therefore, a higher amount of AgNPs may be taken up by phagocytes. When AgNPs of different sizes were applied to macrophage-differentiated human monocyte U937 cells, smaller particles induced a more pronounced decrease in cell viability. In addition, AgNPs enhanced interleukin (IL)-8 secretion. Exposure to AgNPs 4 nm in diameter upregulated intracellular ROS levels. As the induction of IL-8 and intracellular ROS levels was suppressed by N-acetylcysteine treatment, oxidative stress may be involved in the upregulation of IL-8 secretion by AgNPs [21]. Moreover, AgNPs have been reported to increase IL-8 secretion in macrophage-like differentiated THP-1 (dTHP-1) cells [22]. In many cases, exposure to these metal-based nanoparticles increases intracellular ROS levels, activates cellular antioxidant systems such as HO-1, and induces chemokine expression, such as IL-8 [23]. Furthermore, metal-based nanoparticle exposure results in mitochondrial dysfunction, cell membrane damage, and subsequent cell death. In some cases, apoptotic cell death was observed following exposure to metal-base nanoparticles, including AgNPs [24–29]. It has been reported that AgNPs induce apoptosis by upregulating caspase-3 and -9 in rat tracheal epithelial cells [30].

As mentioned above, AgNPs showed various cellular influences. The cellular influences caused by AgNPs depend on physical and chemical properties of AgNPs depending on the use purposes.

The evaluation of cellular responses according to the utilization purpose of the AgNP will be important. In this study, we compared the cytotoxic activity of two types of AgNPs on epithelial-like A549 cells and macrophage-like dTHP-1 cells. One is applied for industrial use. Another one is applied for cosmetic supplementation. The aim of this study is to confirm the cellular effects of AgNP suspensions of different use.

## 2. Materials and Methods

### 2.1. Chemicals

Two types of AgNP water suspension were obtained from Tokuriki Honten (Tokyo, Japan) and Utopia Silver Supplements (Utopia, TX, USA). Detailed information on the AgNP suspensions is presented in Table 1. AgNP water suspensions were serially diluted to the appropriate concentrations in the cell culture medium. Silver nitrate ($AgNO_3$) was purchased from FUJIFILM Wako Pure Chemical Corporation (Osaka, Japan).

**Table 1.** Properties of silver nanoparticles employed in this study.

| Code in This Study | Primary Particle Size (nm) | Included Chemicals in the Suspension | Application | Manufacturer |
|---|---|---|---|---|
| AgNP-A | 28.4 ± 8.5 [1] | Nothing | Cosmetically suppliment | Utopia Silver Supplements |
| AgNP-B | 20 | Nitrogen containing compound | Industrial application | Tokuriki Honten Co., Ltd. |

[1] Primary particle size was reported by Kataoka et al. [31].

### 2.2. Cell Culture

A549 human lung carcinoma cells were purchased from the RIKEN BioResource Center (Tsukuba, Ibaraki, Japan) and cultured in Dulbecco's modified Eagle's medium (DMEM; Gibco, Thermo Fisher Scientific Inc., Waltham, MA, USA) supplemented with 10% heat-

inactivated fetal bovine serum (FBS; HyClone Laboratories, GE Healthcare), 100 units/mL penicillin, 100 μg/mL streptomycin, and 250 ng/mL amphotericin B (antibiotic mixture; Nacalai Tesque Inc., Kyoto, Japan). THP-1 human monocyte cells were obtained from the Japanese Collection of Research Bioresources Cell Bank (Sennan, Japan) and cultured in RPMI1640 medium (Thermo Fisher Scientific Inc.) supplemented with 10% FBS and an antibiotic mixture. The cells were seeded in multi-well plates at $2 \times 10^5$ cells/mL. Then, the cells were incubated for 24 h at 37 °C in an atmosphere containing 5% $CO_2$. Subsequently, to be differentiated into macrophage-like cells, phorbol 12-myristate 13-acetate (PMA; Sigma-Aldrich, St. Louis, MO, USA)-dimethyl sulfoxide solution was added to the culture medium of THP-1 cells at a final concentration of 60 ng/mL, and the cells were further incubated for 48 h. The culture medium was exchanged to the culture medium without PMA three times every day. Thereafter, the culture medium was replaced with an AgNP suspension or AgNO3 solution.

### 2.3. Measurement of Mitochondrial Activity

The cells were seeded in a 96-well plate. After incubation for 24 h, the culture medium was replaced to an AgNP suspension or AgNO3 solution, and the cells were incubated for an additional 24 h. Mitochondrial activity was determined by WST-1 assay using a PreMix WST-1 Cell Proliferation Assay System (Takara Bio Inc., Otsu, Japan). After removing the AgNP suspension or AgNO3 solution, WST-1 reagent was added, and the cells were incubated at 37 °C for 1 h. The incubated sample solution containing WST-1 reagent was transferred to a new 96-well plate, and the optical density was measured at 450 nm using an Infinite F200 PRO microplate reader (Tecan Group Ltd., Männedorf, Switzerland).

### 2.4. Measurement of Intracellular ROS Level

The intracellular ROS level was detected using 2′,7′-dichlorodihydrofluorescein diacetate (DCFH-DA; Sigma-Aldrich). Cells were seeded in a 12-well plate, before the culture medium was exchanged for an AgNP suspension or AgNO3 solution. Cells were incubated for an additional 24 h, and the medium was replaced with serum-free DMEM containing 10 μM DCFH-DA. Next, the cells were incubated for 30 min at 37 °C, washed once with PBS, collected by 0.25% trypsinization, washed once again with PBS, and resuspended in 500 μL PBS. The emission of 2′,7′-dichlorofluorescein (DCF) was measured by a FACSCalibur flow cytometer (Becton, Dickinson and Company, Franklin Lakes, NJ, USA). Data were collected from 5000 gated events.

### 2.5. Determination of IL-8 Level in Culture Medium

To determine IL-8 concentration, cells were seeded in a 24-well plate. The culture medium was removed, and the cells were incubated for an additional 24 h with an AgNP suspension or AgNO3 solution. The level of secreted IL-8 in culture medium was determined by enzyme-linked immunosorbent assay (ELISA) using a Human IL-8 ELISA Ready-SET-Go! kit (Thermo Fisher Scientific Inc.).

### 2.6. Transmission Electron Microscopy (TEM)

Cells were treated with an AgNP suspension at 100 μg/mL in a 35-mm dish and incubated for 24 h. After that, the suspension was removed, and the cells were washed once with PBS. The cells were then fixed with 2% paraformaldehyde and 2% glutaraldehyde in 0.1 mol/L phosphate buffer (PB; pH 7.4) at 4 °C for 30 min. Then, the cells were fixed with 2% glutaraldehyde in 0.1 mol/L PB at 4 °C overnight. After fixation, the samples were washed thrice with 0.1 mol/L PB for 30 min each and postfixed with 2% osmium tetroxide in 0.1 mol/L PB at 4 °C for 1 h. The cell samples were dehydrated in graded ethanol solutions, transferred to a resin (Quetol-812; Nisshin EM Co., Tokyo, Japan), and then polymerized. The polymerized resins were ultra-thin sectioned at 70-nm thickness with a diamond knife using an ultramicrotome (Ultracut UCT; Leica, Vienna, Austria). The cell samples were observed under a transmission electron microscope (JEM-1400Plus;

JEOL Ltd., Tokyo, Japan). The observations were performed by Tokai Electron Microscopy, Inc. (Nagoya, Japan).

### 2.7. Gene Expression

For the determination of gene expression, cells were seeded in 6-well microplates and treated with an AgNP suspension or AgNO$_3$ solution for 24 h. Target gene expression was determined by real-time PCR. Total RNA from cells was purified using an RNeasy mini kit (Qiagen GmbH, Hilden, Germany), and cDNA synthesis was carried out using a High-Capacity cDNA Reverse Transcription kit (Applied Biosystems; Thermo Fisher Scientific, Inc., Waltham, MA, USA). Real-time PCR was conducted using a 7300 Real-Time PCR System (Applied Biosystems), and PCR amplification was performed using a TaqMan® gene expression assay (Applied Biosystems). The human β-actin gene was used as an endogenous control. The TaqMan® probe codes for human β-actin, heme oxygenase-1 (HO-1), IL-8, and metallothionein 2A (MT2A) gene expression assays were Hs99999903_m1, Hs01110250_m1, Hs00174103_m1, and Hs01591333_g1, respectively.

### 2.8. Measurement of Silver Ions Released from AgNPs

The amount of silver ions released from AgNPs was evaluated in DMEM supplemented with 10% FBS and under acidic conditions (20 mM citrate buffer, pH 4.5). AgNP suspensions were diluted in the cell culture medium or citrate buffer at a concentration of 10 μg/mL and then incubated at 37 °C for 24 h. The nanoparticles were separated from the solution via ultrafiltration using Amicon filters (Amicon Ultra Centrifugal Filters Ultracel 10 K; Merck Millipore, Billerica, MA, USA) and centrifuged at 3000× $g$ for 30 min. The filtrate was collected, and the amount of silver ions was measured using inductively coupled plasma-mass spectrometry (ICP-MS; ELAN DRC II; PerkinElmer Life and Analytical Sciences, Shelton, CT, USA). ICP-MS measurements were performed by Sumika Chemical Analysis Service, Ltd. (Osaka, Japan).

### 2.9. Determination of Intracellular Silver Concentration

Cells were seeded in a 6-well plate to determine the intracellular silver ion concentration. Culture medium was replaced with a silver nanoparticle suspension (10 μg/mL), or an AgNO$_3$ solution (100, 500, and 1000 μmol/L) was applied and incubated for an additional 24 h. The suspension or solution was removed, and cells were washed twice with PBS to remove loosely bound silver nanoparticles on the cell surface. Next, cells were collected by 0.25% trypsin treatment, suspended in hypotonic buffer (20 mM Tris, 10 mM NaCl, 3 mM MgCl$_2$ and 0.5 mM dithiothreitol), and incubated for 15 min on ice. The cell lysate was supplemented with Nonidet P-40 and centrifuged at 4 °C, 1000× $g$, for 10 min. The supernatant was collected to measure the cytosolic silver concentration. Silver ion concentration was determined using ICP-MS XSERIES (Thermo Scientific).

### 2.10. Statistical Analysis

Data are expressed as the mean ± standard deviation. Statistical analyses were performed using analysis of variance (ANOVA) and Dunnett's test for multiple comparisons. The calculation method is described in the figure legend.

## 3. Results

### 3.1. Cellular Uptake of AgNPs

The cellular uptake of AgNPs in A549 and dTHP-1 cells following treatment with 10 μg/mL AgNPs was observed by TEM (Figure 1). In AgNP-A-treated THP-1 cells, the cellular uptake of particulate silver was observed. In contrast, in AgNP-B-treated THP-1 cells, fibriform matter was frequently observed in addition to particulate AgNPs. The details of the fibriform matter were unclear. The fibriform matter had a higher electron density than other cell organelles and was observed with AgNPs. Therefore, it may be a binder compound present in the AgNP suspension. The cellular uptake of AgNP-A was

observed in almost all cells in A549 cells. In contrast, the cellular uptake of AgNP-B could not be observed. Intracellular AgNP-B particles were hardly observed in A549 cells.

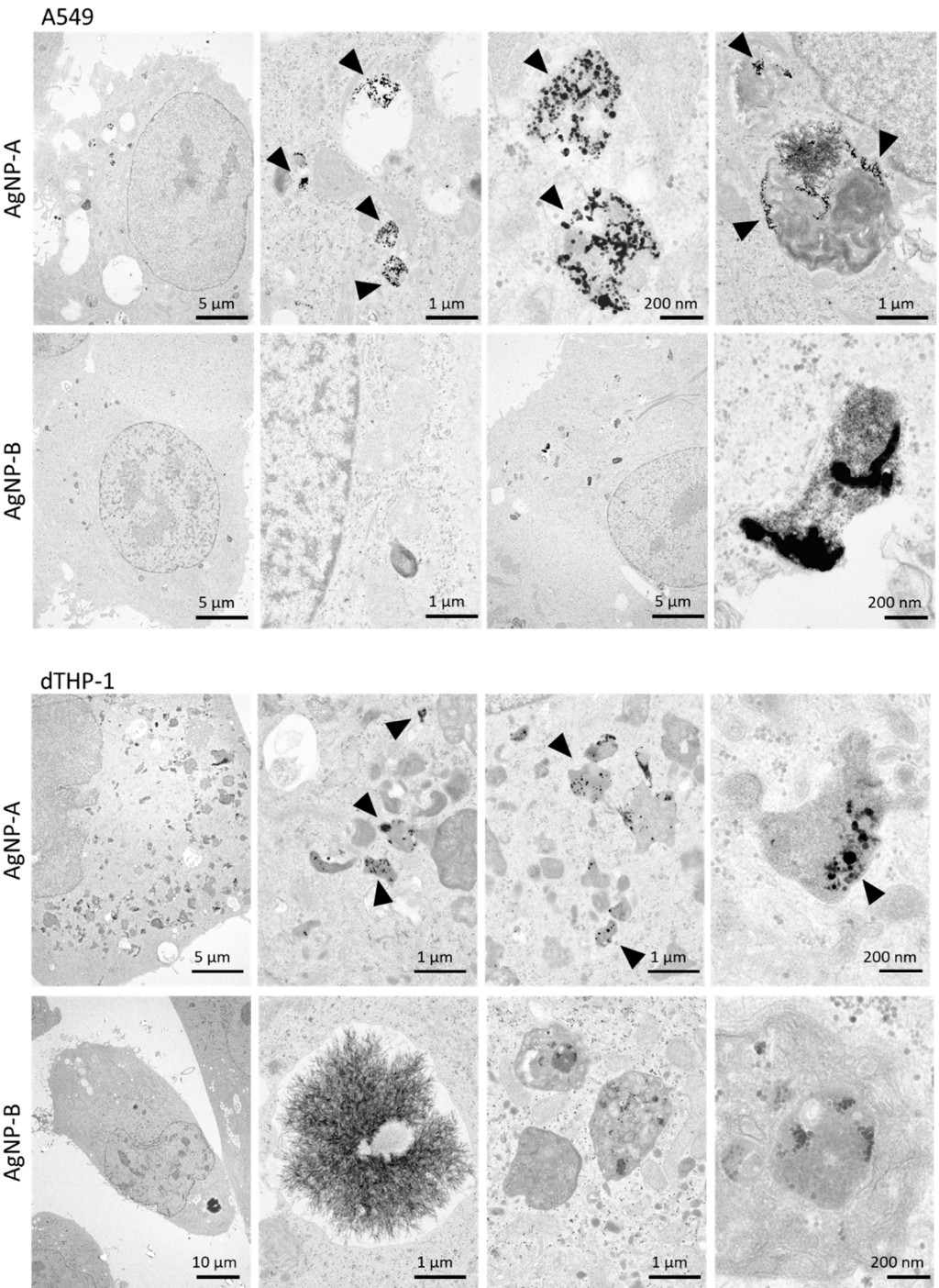

**Figure 1.** TEM observation of AgNP-treated cells. A549 and dTHP-1 cells were exposed to AgNP suspensions for 24 h at 10 μg/mL. Intracellular putative AgNPs were indicated by arrowhead.

### 3.2. Intracellular Silver Levels after Cell Exposure to a Silver Nanoparticle Suspension

The intracellular silver concentration in silver nanoparticle-exposed cells was measured (Figure 2a). In A549 cells, the intracellular silver concentration was upregulated by both AgNP-A and AgNP-B exposure. In dTHP-1 cells, upregulated intracellular silver levels were also observed in AgNP-A- and AgNP-B-exposed cells. MT2A gene expression was remarkably enhanced by exposure to AgNP-A and AgNP-B (Figure 2b). MT2A levels were

1958 and 1350 times higher in AgNP-A- and AgNP-B-exposed A549 cells than in unexposed cells, respectively. MT2A expression was more pronouncedly enhanced in A549 cells than in dTHP-1 cells. On the other hand, the effect of AgNO$_3$ exposure to MT2A gene expression was smaller than Ag nanoparticles. MT2A levels were not affected by AgNO$_3$ exposure at the concentration of 10 μM in both A549 and dTHP-1 cells.

(**a**) Intracellular Ag$^{2+}$ level

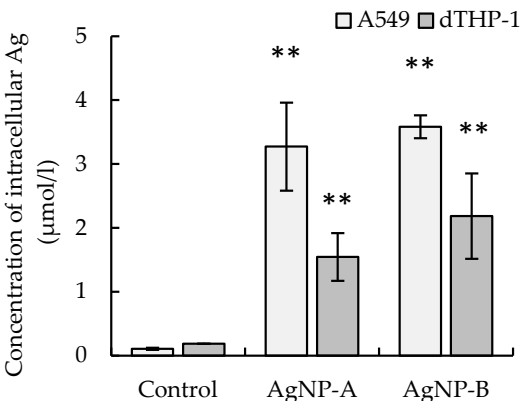

(**b**) MT2A gene expression level

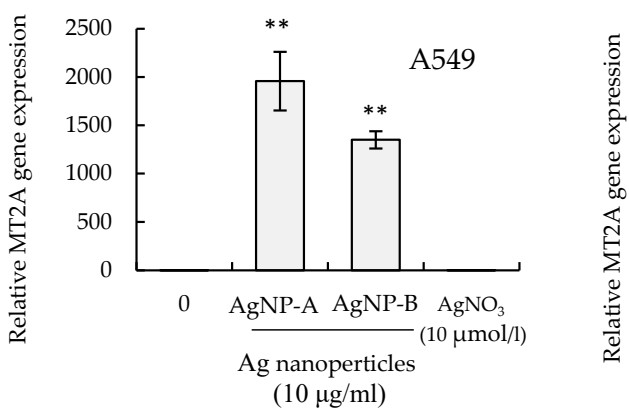
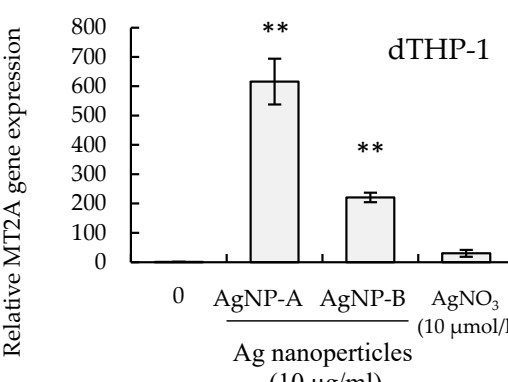

**Figure 2.** Effects of AgNPs on intracellular silver level. (**a**) intracellular silver concentration in silver nanoparticle-exposed cells. A549 and dTHP-1 cells were exposed by silver nanoparticle suspensions for 24 h at concentration of 10 μg/mL. Intracellular silver ion levels were determined by ICP-MS. (**b**) Effect of silver nanoparticles and AgNO$_3$ on MT2A gene expression. A549 and dTHP-1 cells were exposed by silver nanoparticle suspensions for 24 h at concentration of 10 μg/mL. AgNO$_3$ was administered at 10 μmol/L. MT2A gene expression levels were determined by real time-PCR. The MT2A gene expression in standardized untreated cells was 1. ** $p < 0.01$ (vs. unexposed cells, ANOVA and Dunnett's test).

### 3.3. Release of Silver Ions from AgNPs

The concentration of silver ions into the suspension was measured (Table 2). The concentration of silver ions in the DMEM suspension of AgNP-A and AgNP-B after incubation for 24 h at 37 °C was 5.6 and 4.6 μmol/L, respectively. In contrast, a significant silver ion release was observed in the citrate buffer (pH 4.5) suspension of AgNP-A. The concentration of silver ions in the citrate buffer suspensions of AgNP-A and AgNP-B after incubation was 22.2 and 7.4 μmol/L, respectively. The silver concentration of the acidic AgNP-A suspension was three times higher than that of AgNP-B.

**Table 2.** Released silver ion concentratio from silver nanoparticles.

|  |  | Acidic Condition [1] | DMEM with 10% FBS |
|---|---|---|---|
| Rereased $Ag^{2+}$ | AgNP-A | 22.2 | 5.6 |
| ($\mu$mol/L) | AgNP-B | 7.4 | 4.6 |

Ag nanoparticle suspension at concentration of 10 $\mu$g/mL incubated for 24 h at 37 °C. [1] 20 mM citrate buffer, pH 4.5.

### 3.4. Effects of AgNP Suspension and AgNO₃ Solution on Cell Viability

The effect of AgNPs on cell viability in epithelial-like A549 cells and macrophage-like dTHP-1 cells was examined (Figure 3). Cell viability was determined by the WST-1 assay. In a nano-colloidal silver suspension, exposure to 10 $\mu$g/mL AgNP-A significantly decreased cell viability by 9% and 17% in A549 and dTHP-1 cells, respectively. In contrast, AgNP-B did not affect cell viability. AgNO₃ treatment at a concentration of 100 $\mu$g/mL decreased cell viability.

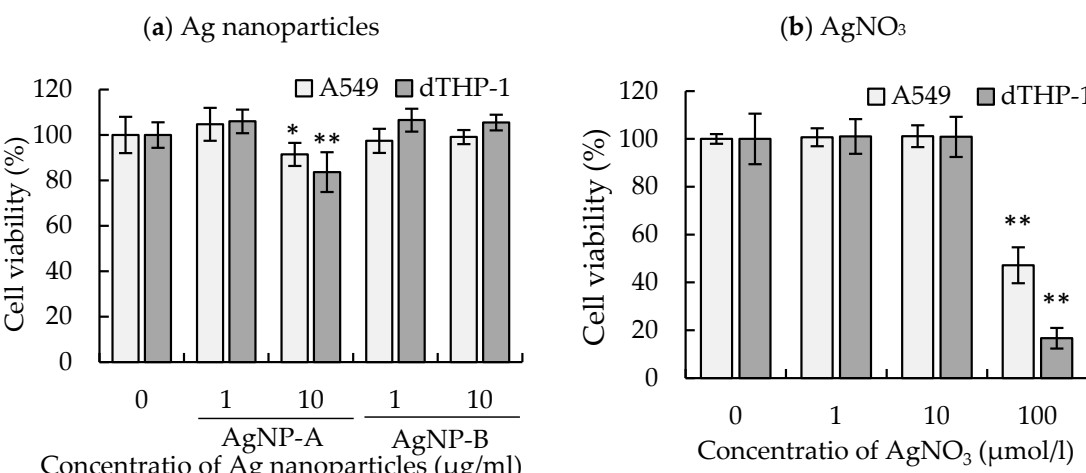

**(a)** Ag nanoparticles   **(b)** AgNO₃

**Figure 3.** Effects of AgNPs (**a**) and AgNO₃ (**b**) on mitochondrial activity. A549 and dTHP-1 cells were treated with AgNP suspensions at 1 and 10 $\mu$g/mL, or with AgNO₃ solution at 1, 10, and 100 $\mu$mol/L. After incubation for 24 h, the WST-1 assay was conducted. The percentage of WST-1 was normalized to that in standardized untreated cells (100%). ** $p < 0.01$, * $p < 0.05$ (vs. untreated cells, ANOVA and Dunnett's test).

### 3.5. Regulation of IL-8 Expression by AgNP Suspension and AgNO₃ Solution

IL-8 secretion was enhanced by exposure to AgNP suspensions (Figure 4). In A549 cells exposed to AgNP-A and AgNP-B (10 $\mu$g/mL), IL-8 expression was 2.7- and 2.1-times higher than that in untreated cells, respectively. IL-8 secretion was markedly enhanced by exposure to AgNPs in dTHP-1 cells. IL-8 levels in the culture supernatant of dTHP-1 cells exposed to 10 $\mu$g/mL AgNP-A and AgNP-B suspensions were 14- and 2.8-times higher than those in untreated cells, respectively. Moreover, the gene expression of IL-8 was enhanced by exposure to 10 $\mu$g/mL AgNP-A and AgNP-B suspensions. The IL-8 gene expression levels were 3.0- and 5.3-times higher in AgNP-A- and AgNP-B-treated A549 cells, respectively. Furthermore, the IL-8 gene expression in AgNP-A- and AgNP-B-treated dTHP-1 cells was 25.8- and 9.7-times higher, respectively. IL-8 secretion was slightly increased by exposure to 10 $\mu$M AgNO₃. Exposure to 10 $\mu$mol/L AgNO₃ did not affect IL-8 gene expression levels.

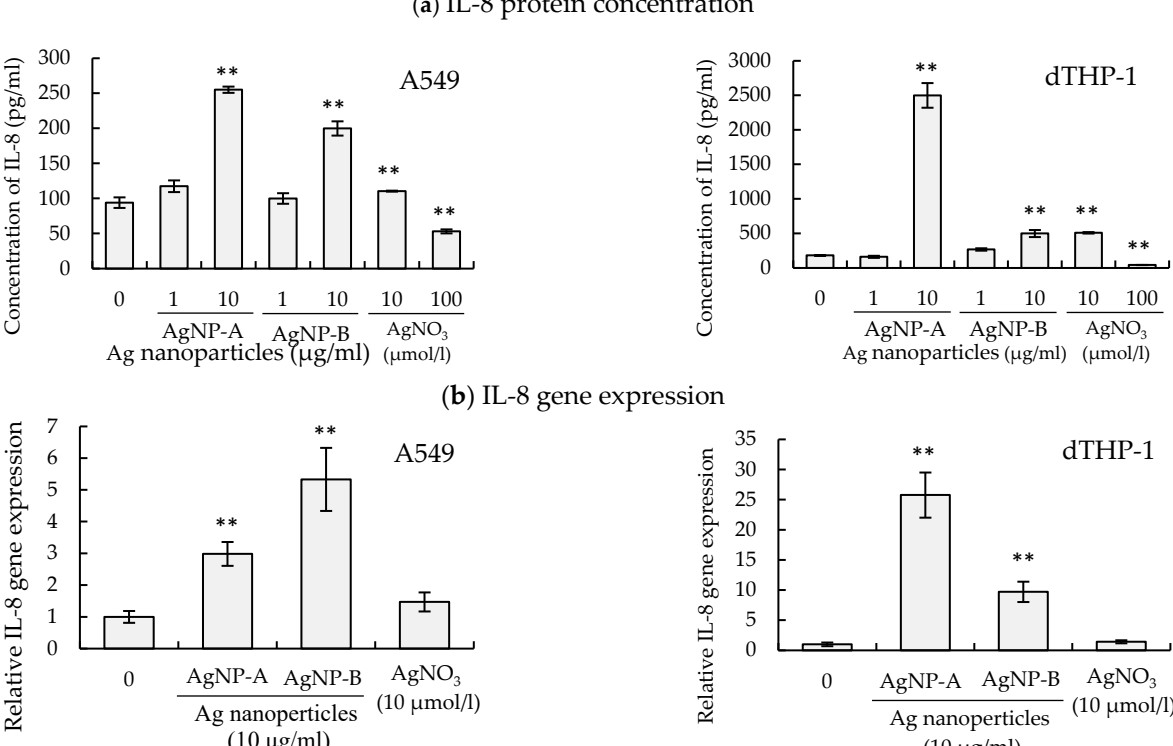

**Figure 4.** Regulation of IL-8 expression by AgNPs and AgNO₃. (**a**) IL-8 protein concentration in culture supernatant. A549 and dTHP-1 cells were exposed to AgNP suspensions at 1 and 10 μg/mL or AgNO₃ at 10 μmol/L. After incubation for 24 h, the culture supernatant was collected, and the level of IL-8 was determined by ELISA. (**b**) IL-8 gene expression. A549 and dTHP-1 cells were treated with AgNP suspensions at 10 μg/mL or AgNO₃ at 10 μmol/L. After incubation for 24 h, IL-8 gene expression was determined by real-time PCR. ** $p < 0.01$ (vs. untreated cells, ANOVA and Dunnett's test).

### 3.6. Effects of AgNP Suspension and AgNO₃ Solution on Oxidative Stress

Intracellular ROS levels were evaluated after exposure to AgNPs for 24 h (Figure 5a). In A549 cells, the intracellular ROS level was increased by exposure to AgNP suspensions. Intracellular ROS levels in AgNP-B-treated cells were approximately 1.5-times higher than those in untreated cells. The intracellular ROS level in AgNP-A-treated cells was 2.6-times higher than that in untreated cells. In dTHP-1 cells, AgNP-B administration increased intracellular ROS level to approximately 1.1-times higher than that in untreated cells. Moreover, intracellular ROS level was 2.2-times higher in AgNP-A-treated cells than in untreated cells. On the contrary, 10 μmol/L AgNO₃ did not increase the intracellular ROS level in either A549 or dTHP-1 cells. Furthermore, the gene expression of HO-1, an important oxidative stress response protein, in cells exposed to AgNPs was evaluated as an oxidative stress marker (Figure 5b). HO-1 levels were induced by exposure to 10 μg/mL AgNP-A and AgNP-B suspensions. The HO-1 levels were 7.9- and 10.1-times higher in AgNP-A- and AgNP-B-treated A549 cells, respectively. Furthermore, HO-1 gene expression in AgNP-A- and AgNP-B-treated dTHP-1 cells was 35.3- and 13.9-times higher, respectively. In addition, AgNO₃ exposure at 10 μmol/L did not increase the HO-1 level in either A549 or dTHP-1 cells.

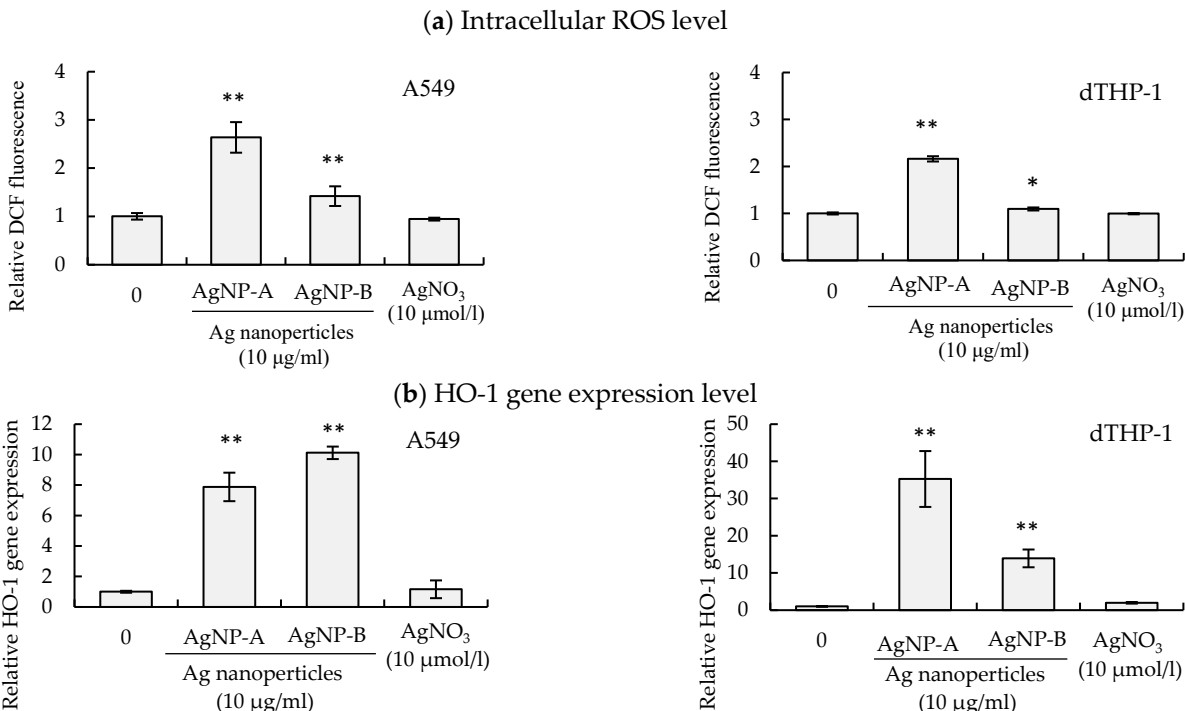

**Figure 5.** Effects of AgNPs and AgNO$_3$ on intracellular oxidative stress. (**a**) Intracellular ROS levels. A549 and dTHP-1 cells were treated with AgNP suspensions at 10 μg/mL or AgNO$_3$ at 10 μmol/L. After incubation for 24 h, intracellular ROS levels were measured by the DCFH assay. The value of DCF fluorescence in standardized untreated cells was 1. (**b**) HO-1 gene expression. A549 and dTHP-1 cells were treated with AgNP suspensions at 10 μg/mL or AgNO$_3$ at 10 μmol/L. After incubation for 24 h, HO-1 gene expression was determined by real-time PCR. ** $p < 0.01$, * $p < 0.05$ (vs. untreated cells, ANOVA and Dunnett's test).

## 4. Discussion

In many cases, cellular uptake and intracellular metal ion release are important factors for the cytotoxicity of metal-based nanoparticles [18,21]. In the present study, basic cellular responses caused by two silver nanoparticle suspensions were similar in both A549 and dTHP-1 cells. The exposure of silver nanoparticle suspensions caused an enhancement of IL-8 expression, intracellular oxidative stress, and MT2A gene expression. In contrast, cell viability was hardly affected by silver nanoparticle suspensions. Cellular responses tended to be stronger by AgNP-A than AgNP-B. When the AgNP suspension was diluted to 10 μg/mL in the culture medium, the extracellular Ag$^{2+}$ concentration was approximately 5–6 μmol/L. However, the efficacies of 10 μmol/L AgNO$_3$ solution in inducing IL-8 expression and intracellular oxidative stress were weaker than those of AgNP suspensions. AgNO$_3$ solution at 10 μmol/L did not affect the gene expression of IL-8 or cellular oxidative stress. AgNO$_3$ hardly showed any cellular effect at a concentration of 10 μmol/L. These results suggest that extracellular dissolved Ag$^{2+}$ is not a major factor in the cellular responses caused by AgNPs. In particular, AgNP-A were more soluble under acidic conditions. Therefore, it is possible that intracellular AgNPs release silver ions at the phagolysosome, which is acidic. The cellular uptake of AgNP-A was observed in both A549 and dTHP-1 cells. In contrast, the cellular uptake of AgNP-B was hardly observed in A549 cells. However, interestingly, the intracellular Ag$^{2+}$ concentration increased in both AgNP-A- and AgNP-B-treated A549 cells. Moreover, intracellular Ag$^{2+}$ levels were higher in A549 cells than in dTHP-1 cells. The intracellular silver concentration in A549 and dTHP-1 cells was approximately 3.4 and 1.9 μmol/L, respectively. Furthermore, the gene expression of MT2A was remarkably enhanced by AgNP suspension. MT2A expression was also enhanced in AgNP-B-treated A549 cells. Therefore, there is a possibility that intracellular Ag$^{2+}$ levels were increased in

AgNP-B-treated A549 cells. However, intracellular AgNP-B were not detected. Although intracellular $Ag^{2+}$ may be important for cellular responses caused by silver nanoparticles, the mechanism of intracellular $Ag^{2+}$ induction is unclear. The reason for the increase in intracellular $Ag^{2+}$ levels in AgNP-B-treated A549 cells is unknown.

In any of these cases, both AgNP suspensions elevated intracellular ROS levels and IL-8 expression. These cellular responses have previously been observed in cells exposed to cytotoxic metal-based nanoparticles, such as ZnO and CuO nanoparticles [24]. However, the effect of AgNP exposure on mitochondrial activity was weak. The cytotoxic ZnO and CuO nanoparticles can decrease mitochondrial activity at a concentration of 10 µg/mL, and one of the two types of AgNPs used in this study also reduced mitochondrial activity. AgNP-A decreased mitochondrial activity in both A549 and dTHP-1 cells. Although the two types of AgNPs induced similar cellular responses, the strength of the responses depended on the type of AgNP suspension. The AgNPs used in this study decreased mitochondrial activity and increased intracellular oxidative stress and IL-8 expression. However, whether these cellular effects are associated with intracellular silver ion release from AgNPs requires further study. In A549 cells, the intracellular silver concentration was significantly higher in AgNP-A- and AgNP-B-treated cells than in untreated cells. The AgNP-induced increase in IL-8 and HO-1 expression was higher in dTHP-1 cells than in A549 cells. The gene expression levels of IL-8 in AgNP-A- and AgNP-B-treated dTHP-1 cells were 8.6- and 1.8-times higher than those in A549 cells, respectively. The gene expression levels of HO-1 in AgNP-A- and AgNP-B-treated dTHP-1 cells were 4.5- and 1.4-times higher than those in A549 cells, respectively. Furthermore, the gene expression of MT2A in AgNP-A- and AgNP-B-treated A549 cells was 3.1- and 6.1-times higher, respectively, than that in dTHP-1 cells. MT2A gene expression was remarkably enhanced in A549 cells following exposure to AgNP suspensions. In other words, the increase in IL-8 and HO-1 gene expression was dTHP-1 cells > A549 cells, whereas the increase in MT2A gene expression was A549 cells > dTHP-1 cells. It should be noted that MTA2 binds to and detoxifies heavy metals such as cadmium. Thus, it is possible that the silver ions released from intracellular AgNPs were inactivated by MT2A.

Therefore, intracellular silver concentration is an important factor in the efficacy of AgNPs in inducing IL-8 and HO-1 expression. The increase in intracellular silver concentration may be caused by the cellular uptake and subsequent intracellular dissolution of AgNPs. However, the reason why intracellular AgNP-B were undetectable is unclear. One possibility is that intracellular AgNP-B were rapid and completely dissolved. Although the solubility of AgNP-B under acidic conditions was lower than that of AgNP-A, AgNP-B contained a nitrogen-containing compound as a binder. Thus, the actual mass of AgNPs in the AgNP-B suspension may be smaller. However, we do not yet have any evidence.

In fact, the findings of MT2A expression and intracellular silver levels suggested that intracellular AgNP-B were more soluble than AgNP-A. Therefore, binder compounds may have affected the intracellular dissolution of AgNP-B. The cellular uptake and intracellular dissolution of AgNPs are certainly causes of their cellular effects. On the other hand, intracellular dissolution is not the sole cause of the cellular effect caused by AgNPs. Other properties of AgNPs, such as surface activity and figure, may also be important. Understanding of these properties of AgNPs is important for the utilization of AgNPs (Figure 6).

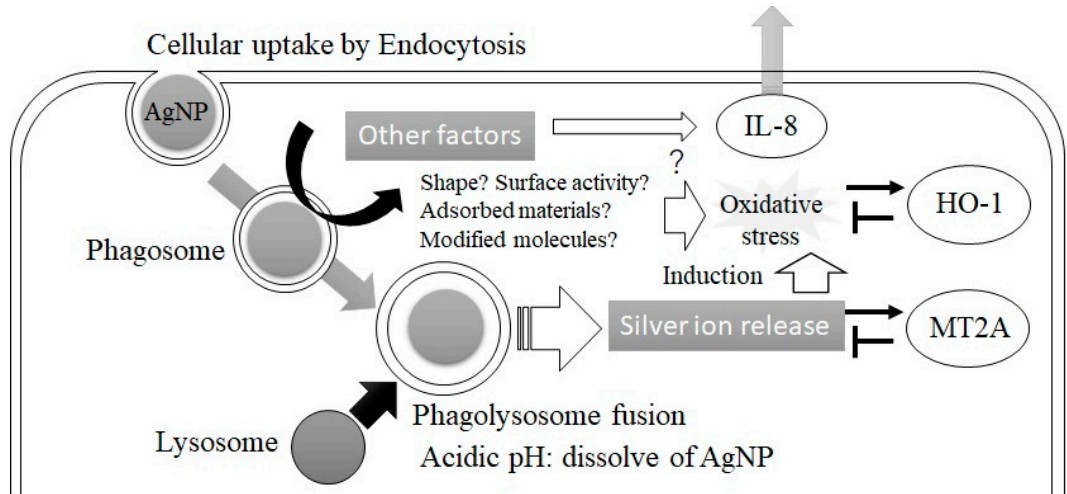

**Figure 6.** Summary of cellular effects caused by AgNP in this study.

In conclusion, AgNPs induced IL-8 and HO-1 expression, but cytotoxicity was weak. Moreover, increased intracellular silver concentrations contributed to the elevation of IL-8 and HO-1 expression. The increase of intracellular $Ag^{2+}$ level caused by silver nanoparticle suspensions may be important for these cellular responses.

**Author Contributions:** Conceptualization, M.H. and K.S.; methodology, M.H.; investigation, K.S.; data curation, M.H. and K.S.; writing—original draft preparation, M.H.; writing—review and editing, M.H., K.S. and S.K.; supervision, M.H. and S.K. All authors have read and agreed to the published version of the manuscript.

**Funding:** This research received no external funding.

**Institutional Review Board Statement:** Not applicable.

**Informed Consent Statement:** Not applicable.

**Data Availability Statement:** All data are presented in this article in the form of figures and tables.

**Conflicts of Interest:** The authors declare no conflict of interest.

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
