# Peer review of "Cellular Effects of Silver Nanoparticle Suspensions on Lung Epithelial Cells and Macrophages"

_applsci, doi:10.3390/app12073554_

Round 1

Reviewer 1 Report

The presented work discusses the cellular effects of silver nanoparticles. I would like to congratulate the authors on a well-written manuscript and, in my eyes, thoroughly conducted experiments. Nevertheless, I have to suggest a minor revision, as the novelty of the work is not entirely clear in its present version. The topic of nanotoxicity of Ag nanoparticles is a well-studied one and various aspects have been investigated extensively. If the authors can convince the readers of the novelty of this work and, importantly, also its implications on the research field and future studies, I am willing to suggest an acceptance of this manuscript.

Furthermore, I suggest a discussion of the following minor remarks:

  1. I would show the TEMs first as they explain a lot of the observed effects.
  2. You describe and discuss the amount of dissolved Ag. Did you also measure the Ag concentration in the supernatants both in form of Ag nanoparticles as well as Ag ions? This would allow you to completely close the mass balance of your system and get more insights into the dissolution processes?

Author Response

We are most grateful to you and the reviewer1 for the helpful comments on the original version of our manuscript.

Response 1) Order of figures was changed. The results section was reconstructed.

Response 2) Unfortunately, we did not measure the Ag concentration in the supernatants. Following sentence was added to the discussion section: “Cellular uptake and intracellular dissolution of AgNPs are certainly cause of their cellular effects. On the other hand, intracellular dissolution is not sole cause of cellular effect caused by AgNPs. Other properties of AgNPs such as surface activity and figure may be also important. Understanding of these properties of AgNPs are important for utilization of AgNPs.” And summarized figure of the findings was also added.

Reviewer 2 Report

The authors bring out very informative research work, but few minor comments need to be clarified before acceptance.

Line no 2: delete two kinds of

Line no 11: What this means-  Although several types of AgNPs (Give clear frame).

As mentioned, Two types of AgNP water suspension were obtained from Tokuriki Honten and Uto- 81pia Silver Supplements (Utopia, TX, USA). Had authors tried the lab synthesized nanoparticles, if yes, what was the difference?

Author mentioned the different concentrations for AgNP figure 1 caption but not reflecting in Figure 1, modify the image.  

AgNO3 is well known, why author added the data here?

Improve the TEM images to show the NP?

In discussion part, author mentioned some research work is unclear based on their findings, the question is all these activities by the nanoparticles or by the surface moiety?

Also, author need to suggest diagrammatic representation of their findings?

This manuscript shows the important scientific progress in content, some literature with similar line of work should be cited for broader readership, such as

International Journal of Nanomedicine 2021:16 Pages 7711—7726

Spectrochimica Acta Part A: Molecular and Biomolecular Spectroscopy (2013). 114:144-147.

RSCAdv., 2021, 11, 9880-9893

Bioorganic Chemistry 2021, 107, 104626

Biomolecules 2021, 11, 385

Nanomaterials 2021, 11, 322

Author Response

We are most grateful to you and the reviewer2 for the helpful comments on the original version of our manuscript.

Response 1) The sentence “two kinds of” was deleted.
Response 2) The sentence was revised as follows: “Generally, AgNPs have shown cytotoxicity such as cell membrane damage and oxidative stress induction,”
Response 3) We did not synthesize Ag nanoparticles ourself. In the present study, we evaluated only commercially available Ag nanoparticles for different use application: cosmetics and industry. Additionally, we didn't have equipment for synthesis.
Response 4) The caption of Fig.1 was corrected.
Response 5) We evaluated cellular effect of AgNO3 to discuss effect of extracellular Ag ion by same experiment system with Ag nanoparticles.
Response 6) Intracellular putative AgNPs were indicated by arrowhead.
Response 7) Following sentence was added to the discussion section: “Cellular uptake and intracellular dissolution of AgNPs are certainly cause of their cellular effects. On the other hand, intracellular dissolution is not sole cause of cellular effect caused by AgNPs. Other properties of AgNPs such as surface activity and figure may be also important. Understanding of these properties of AgNPs are important for utilization of AgNPs.” And summarized figure of the findings was also added.
Response 8) Some suggested literature were cited.
